# Should Age-Dependent Absolute Risk Thresholds Be Used for Risk Stratification in Risk-Stratified Breast Cancer Screening?

**DOI:** 10.3390/jpm11090916

**Published:** 2021-09-15

**Authors:** Nora Pashayan, Antonis C. Antoniou, Andrew Lee, Michael Wolfson, Jocelyne Chiquette, Laurence Eloy, Andrea Eisen, Tracy L. Stockley, Hermann Nabi, Jennifer D. Brooks, Michel Dorval, Douglas F. Easton, Bartha Maria Knoppers, Anna M. Chiarelli, Jacques Simard

**Affiliations:** 1Department of Applied Health Research, University College London, London WC1E 7HB, UK; 2Centre for Cancer Genetic Epidemiology, Department of Public Health & Primary Care, School of Clinical Medicine, University of Cambridge, Cambridge CB1 8RN, UK; aca20@medschl.cam.ac.uk (A.C.A.); ajl65@medschl.cam.ac.uk (A.L.); dfe20@medschl.cam.ac.uk (D.F.E.); 3School of Epidemiology and Public Health, University of Ottawa, Ottawa, ON K1G 5Z3, Canada; michael.wolfson@uottawa.ca; 4Department of Family Medicine and Emergency Medicine, Faculty of Medicine, Université Laval, Quebec City, QC G1V 4G2, Canada; jocelyne.chiquette.med@ssss.gouv.qc.ca; 5CHU de Québec-Université Laval Research Centre, Quebec City, QC G1S 4L8, Canada; Hermann.Nabi@crchudequebec.ulaval.ca (H.N.); Michel.Dorval@crchudequebec.ulaval.ca (M.D.); jacques.simard@crchudequebec.ulaval.ca (J.S.); 6Quebec Cancer Program, Ministère de la Santé et des Services Sociaux, Quebec City, QC G1S 2M1, Canada; laurence.eloy@msss.gouv.qc.ca; 7Sunnybrook Health Science Centre, Toronto, ON M4N 3M5, Canada; andrea.eisen@sunnybrook.ca; 8Division Clinical Laboratory Genetics, Laboratory Medicine Program, University Health Network, Toronto, ON M5G 2C4, Canada; tracy.stockley@uhn.ca; 9Department of Laboratory Medicine and Pathobiology, University of Toronto, Toronto, ON M5S 1A8, Canada; 10Department of Social and Preventive Medicine, Faculty of Medicine, Université Laval, Quebec City, QC G1V 0A6, Canada; 11Dalla Lana School of Public Health, University of Toronto, Toronto, ON M5S 1A1, Canada; jennifer.brooks@utoronto.ca; 12Faculty of Pharmacy, Université Laval, Quebec City, QC G1V 0A6, Canada; 13CISSS de Chaudière-Appalaches Research Center, Lévis, QC G6V 3Z1, Canada; 14Centre of Genomics and Policy, McGill University, Montreal, QC H3A 0G1, Canada; bartha.knoppers@mcgill.ca; 15Ontario Health, Cancer Care Ontario, Toronto, ON M5G 2L3, Canada; anna.chiarelli@ontariohealth.ca; 16Department of Molecular Medicine, Faculty of Medicine, Université Laval, Québec City, QC G1V 0A6, Canada

**Keywords:** absolute risk, remaining lifetime risk, risk threshold, risk-stratified screening, misclassification

## Abstract

In risk-stratified cancer screening, multiple risk factors are incorporated into the risk assessment. An individual’s estimated absolute cancer risk is linked to risk categories with tailored screening recommendations for each risk category. Absolute risk, expressed as either remaining lifetime risk or shorter-term (five- or ten-year) risk, is estimated from the age at assessment. These risk estimates vary by age; however, some clinical guidelines (e.g., enhanced breast cancer surveillance guidelines) and ongoing personalised breast screening trials, stratify women based on absolute risk thresholds that do not vary by age. We examine an alternative approach in which the risk thresholds used for risk stratification vary by age and consider the implications of using age-independent risk thresholds on risk stratification. We demonstrate that using an age-independent remaining lifetime risk threshold approach could identify high-risk younger women but would miss high-risk older women, whereas an age-independent 5-year or 10-year absolute risk threshold could miss high-risk younger women and classify lower-risk older women as high risk. With risk misclassification, women with an equivalent risk level would be offered a different screening plan. To mitigate these problems, age-dependent absolute risk thresholds should be used to inform risk stratification.

## 1. Introduction

Risk-stratified screening approaches have emerged as promising strategies to improve the efficiency of cancer screening programmes and to reduce their adverse consequences [1,2,3]. A risk-stratified screening program would involve individualised risk assessment based on a range of risk factors (e.g., genetic, lifestyle, hormonal, reproductive), partitioning population risk into several risk categories, assigning individuals to specific risk categories, and tailoring screening recommendations to each risk category [4]. Risk thresholds for risk stratification are the key to translating the results of individualised risk assessment into screening strategies [5].

Several breast cancer risk assessment models are available. These models estimate a woman’s absolute risk of developing breast cancer, either over a fixed time horizon (e.g., five or ten years) or over her remaining lifetime (e.g., from current age to age 80 or 85) [6,7,8,9].

Clinical guidelines in the US [10,11] and in Canada [12] recommend enhanced breast screening (annual mammogram and adjunctive breast MRI from young age) of women with a lifetime risk for breast cancer of 20% (US)—25% (Canada) or over, as assessed by risk models such as the Tyrer–Cuzick [6] or Breast and Ovarian Analysis of Disease Incidence of Carrier Estimation Algorithm (BOADICEA) [7] risk models. However, when a woman is referred to a high-risk clinic, her risk is assessed from her current age; hence, her remaining lifetime risk is being estimated rather than her lifetime risk. The latter is an estimate of risk from a fixed index age to truncated upper age, e.g., age 20 to 80 years [7]. Whilst remaining lifetime risk of breast cancer decreases with age (Appendix A), the risk stratification is based on an absolute risk threshold that does not vary by age.

Five-year absolute risk thresholds are also used by clinical guidelines [10] and by ongoing personalised breast cancer screening randomised trials [13,14]. For example, the US National Comprehensive Cancer Network (NCCN) guidelines recommend annual mammograms for women over the age of 35 whose 5-year risk of invasive breast cancer, assessed by the modified Gail model, is 1.7% or higher [10]. In the My Personalised Breast Screening (MyPeBS) trial, women aged 40 to 70 years recruited to the risk-based screening arm are stratified into four risk-groups based on 5-year risk estimates: <1% (low risk), 1–1.66% (average risk), 1.67–5.99% (high risk), and ≥6% (very high risk), with the offer of 4-yearly, 2-yearly, annual mammography, and annual mammography with adjunctive MRI, respectively [13]. The 5-year risk is estimated from the age at recruitment to the trial. Whilst 5-year absolute risk of breast cancer increases with age (Appendix A), risk stratification is based on absolute risk thresholds that do not vary by age.

In each of these examples, the risk thresholds do not take into account a woman’s age at the time of risk assessment. However, since the absolute risk of breast cancer varies substantially by age, this will have a substantial influence on the proportion of women falling into each risk category. Here, we examine an alternative approach in which risk thresholds vary by age, consider the implications of using age-independent risk thresholds on risk stratification, and present age-dependent absolute risk thresholds for tailoring screening recommendations in Canada.

## 2. Materials and Methods

We defined age-dependent risk thresholds based on *relative risk* of breast cancer, relative to the population average. We calculated age-dependent risk thresholds (*risk_threshold_* (*t*) at age *t*) that are equivalent to those for a 30-year-old woman having a specified absolute risk, with ‘equivalence’ defined as having the same relative risk (*RR*) (relative to the population age-specific average absolute risk, *risk_pop_*(*t*)).
(1)$$RR=\frac{\log\left(1.0-risk\left(t\right)\right)}{\log\left(1.0-risk_{pop}\left(t\right)\right)}$$

### 2.1. Calculating Age-Dependent Absolute Risk Thresholds

In order to preserve the general intent of the extant guidelines, we considered three remaining lifetime risk categories, all from age 30 to 80 years, of <15%, 15–24% and ≥25% as ‘average’, ‘higher than average’, and ‘high’ risk, respectively. The high-risk threshold from age 30 to 80 corresponds to the definition used by the High-Risk Ontario Breast Screening Program [15]. We calculated age-conditional absolute risk (*risk_pop_*), 5- and 10-year absolute risk and remaining lifetime risk to age 80 for women in Canada of varying ages 30 to 79 years (Appendix A), using DevCan software (version 6.7.6 US National Cancer Institute, SA) [16] and data from Statistics Canada [17] on breast cancer incidence, mortality from breast cancer, and mortality from other causes between 2012 to 2016.

A 30-year-old woman in Canada has a remaining lifetime risk of breast cancer of 10.1%. The 25% and 15% remaining lifetime risk thresholds at age 30 are equivalent to relative risks of 2.7 and 1.5, respectively. Given the age-specific population absolute risk and the relative risk, we calculated age-specific absolute risk thresholds as
(2)$$risk_{threshold}\left(t\right)=1.0-{\left(1-risk_{pop}\left(t\right)\right)}^{RR_{}}$$

In a sensitivity analysis, we calculated 5- and 10-year absolute risk thresholds equivalent to those for a 40-year-old woman having remaining lifetime risk of 25% and 15%.

### 2.2. Estimating the Proportion at High Risk

The overall discriminatory ability of a breast cancer risk prediction model that incorporates a polygenic risk score (based on 313 breast cancer susceptibility variants) and lifestyle, hormonal, and reproductive risk factors is estimated to have an Area Under the Curve (AUC) of 0.65 [18]. Assuming a log-additive interaction between the risk factors, the distribution of risk in the population is log-normal on a relative risk scale [19] with an estimated variance ($\sigma^{2}$) of 0.30 [20]. We set the mean as –$\sigma^{2}$/2, so that the mean relative risk in the population equals to unity [21]. Given the mean and variance of the log-normal relative risk distribution, we calculated the proportion of women that would have an absolute risk (or relative risk) greater than a given absolute risk threshold.

## 3. Results

### 3.1. Age-Dependent Absolute Risk Thresholds

Table 1 presents the 10-year absolute risk thresholds that would classify 40- to 69-year-old women in Canada into three risk categories: ‘average’, ‘higher than average’, and ‘high’ risk. The risk categories are equivalent to remaining lifetime risk (from age 30 to 80) of <15%, 15–24%, and ≥25%, respectively. A 40-year-old woman with a 10-year absolute risk estimate of 3.7% would be classified as high risk, whereas a 60-year-old woman with the same risk estimate would be classified as average risk.

### 3.2. Implications on Risk Stratification

Figure 1 shows the age-specific absolute risk thresholds based on remaining lifetime risk (Figure 1a) and 10-year absolute risk (Figure 1b) of breast cancer incidents, using thresholds equivalent to a remaining lifetime risk at age 30 of 25% (high risk) and 15% (higher than average risk). Remaining lifetime risk decreases with age. A remaining lifetime risk threshold of 25% at age 30 years would be equivalent to remaining lifetime risk thresholds of 20% at age 55 and 14% at age 65. If the threshold for high risk is kept constant with age (25% remaining lifetime risk threshold applied for all ages), then a woman, for example, aged 65 years, with a remaining lifetime risk of 14%, would be classified as average risk, and then would not receive the enhanced screening recommended for high-risk women. An age-independent remaining lifetime risk threshold could identify high-risk younger women but would miss high-risk older women.

Ten-year absolute risk thresholds of 1.1% and 3.7% at ages 30 and 40 would be equivalent to remaining lifetime risk of 25% at ages 30 and 40, respectively. An age-independent 10-year absolute risk threshold (1.1% or 3.7% 10-year absolute risk threshold applied for all ages) could miss high-risk younger women and classify lower-risk older women as at high risk (Figure 1b). Similarly, an age-independent 5-year absolute risk threshold could under-classify younger women and over-classify older women (Appendix A).

Table 2 shows the age-specific relative risk associated with different absolute risk thresholds. Based on Ontario high-risk breast screening guidelines a woman 30 years of age with an RR at or above 2.7, compared to the population average risk, would be considered at high risk. With an age-independent high-risk threshold, based on remaining lifetime risk, a woman, for example, at age 65 would need to have an RR of ≥5.2 to be high-risk, whereas based on 10-year absolute risk, with an RR of ≥0.3 (i.e., less than the population average risk) would be classified as high risk.

Figure 2 presents the proportion of women in Canada age 30 to 69 years considered at high risk when the log-normal distribution of risk in the population has a variance of 0.30, and a 10-year absolute risk threshold for high risk is equivalent to a remaining lifetime risk of 25% at age 40. With an age-dependent absolute risk threshold, even though the absolute risk threshold increases with age, the same proportion of women, 1.8% of women, of any age from age 30 to 69 would be classified as high risk, whereas with age-independent absolute risk threshold, 14.4% of the women, with fewer younger women and many more older women (over 30% of women 60 years or over), would be classified as high risk.

## 4. Discussion

Using an age-independent remaining lifetime risk threshold approach could identify high-risk younger women but would miss high-risk older women, whereas an age-independent 10-year or 5-year absolute risk threshold could miss high-risk younger women and classify lower-risk older women as high risk. Using an age-dependent absolute risk threshold for risk stratification, women with the same absolute risk level but at different ages could receive different screening recommendations. For example, a 40-year-old woman with a 10-year absolute risk of breast cancer of 3.7% would be recommended enhanced screening, whereas a 60-year-old woman with the same risk level would not.

With risk misclassification, women with equivalent risk levels would be offered a different screening plan. An average-risk woman, when classified as high risk, would unnecessarily receive enhanced screening, annual mammograms, and adjunctive breast MRI. This potentially could increase the likelihood of experiencing adverse consequences of screening, false screening findings and overdiagnosis, as well as of inefficient use of healthcare resources, whereas a high-risk woman missing being classified as high risk would miss the benefits of enhanced screening.

If risk is assessed from current age, and the age at assessment varies between individuals, then risk stratification with risk thresholds that are age-dependent should be used to mitigate the problems of misclassification and inequity in provision of the screening services. An alternative is to compute risk from a fixed index age or have a one-off risk assessment at a particular age only in order to use age-independent absolute risk thresholds. For example, the UK National Institute for Health and Clinical Care Excellence (NICE) guidelines define breast cancer risk categories based on remaining lifetime risk from a fixed index age of 20 to 80 years [22].

In the Personalized Risk Assessment for the Prevention and Early Detection of Breast Cancer: Integration and Implementation (PERSPECTIVE I&I) study currently underway in Canada, women aged 40 to 69 years are invited for risk assessment, and a screening plan is proposed based on the women’s risk category, age at risk assessment, and provincial breast screening guidelines [23]. PERSPECTIVE I&I is using age-specific 10-year risk thresholds (Table 1). NICE guidelines also use age-group-specific 10-year risk categories [22]. The 5- or 10-year absolute risk estimates are better supported by evidence from empirical evaluation [24,25], less susceptible to changes in incidence and mortality rates over time [26], better capture risk factors that change over time, and can better inform decisions on screening start or stop ages.

In this analysis we have calculated age-dependent absolute risk thresholds equivalent to the thresholds recommended by the clinical guidelines. The relative risk threshold approach is a pragmatic approach to set age-dependent absolute risk thresholds. However, evidence from empirical data is still needed to determine the absolute risk thresholds to be used in risk-stratified breast screening programmes. In addition, evidence is needed on whether an age-dependent, compared to age-independent, absolute risk threshold would improve screening outcomes, such as reducing breast cancer death and overdiagnosis, and improve the cost-effectiveness of the screening programme. Ongoing studies, such as the PERSPECTIVE I&I, will offer evidence on the optimal risk thresholds to improve the benefit to harm ratio and the cost-effectiveness of risk-stratified breast cancer screening programmes, the clinical utility, and feasibility of implementing age-dependent absolute risk thresholds for risk-stratification in breast cancer screening.

## Figures and Tables

**Figure 1 jpm-11-00916-f001:**
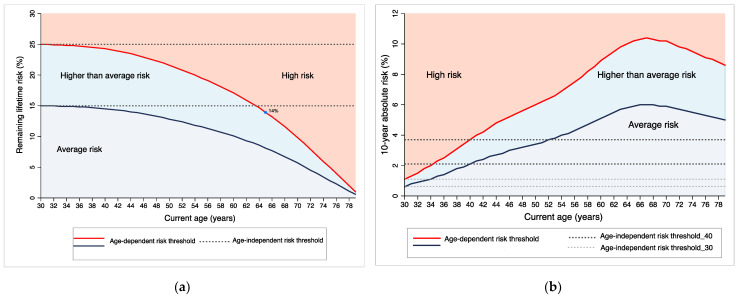
Age-dependent risk thresholds based on (**a**) remaining lifetime risk (from current age to age 80 years) and (**b**) ten-year absolute risk for breast cancer in women in Canada (2012–2016), where risk thresholds for the ‘high risk’ (red line) and ‘higher than average’ (blue line) risk categories have been set at remaining lifetime risks of 25% and 15%, respectively, if determined at age 30 (in (**a**)) and age 40 (in (**b**)). Risk categories: high risk (red line and above), higher than average risk (between the blue line and the red lines), and average risk (below the blue line). Grey dotted lines represent age-independent risk thresholds corresponding to 25% and 15% remaining lifetime risks (from age 30 to 80) (**a**) and 10-year absolute risks of 3.7% and 2.1% when assessed at age 40 are equivalent to 25% and 15% remaining lifetime risks at age 40 (**b**). Light grey dotted lines in (**b**) represent 10-year absolute risks of 1.1% and 0.63% when assessed at age 30, which are equivalent to 25% and 15% remaining lifetime risks also at age 30, respectively.

**Figure 2 jpm-11-00916-f002:**
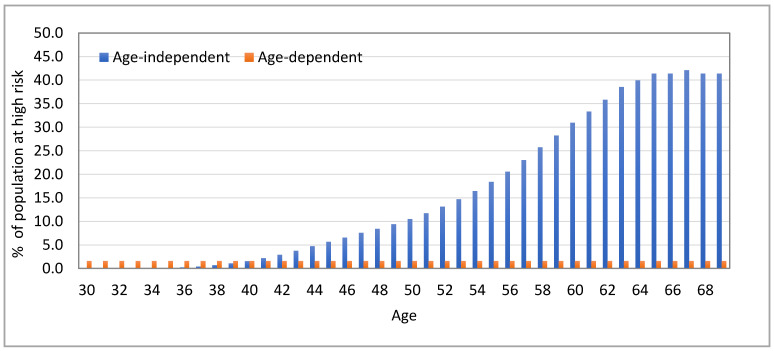
Proportion of women in Canada considered at high risk of breast cancer, considering age-dependent and age-independent 10-year absolute risk thresholds equivalent to a remaining lifetime risk of 25% from age 40.

**Table 1 jpm-11-00916-t001:** Age-specific 10-year absolute risk thresholds for the risk categories—average, higher than average, and high risk for women in Canada (2012–2016).

	Average Risk	Higher Than Average Risk	High Risk
Age	10-Year Absolute Risk %	10-Year Absolute Risk %	10-Year Absolute Risk %
40	[0, 2.0)	[2.0, 3.6)	[3.6, )
41	[0, 2.2)	[2.2, 3.8)	[3.8, )
42	[0, 2.3)	[2.3, 4.1)	[4.1, )
43	[0, 2.5)	[2.5, 4.3)	[4.3, )
44	[0, 2.6)	[2.6, 4.6)	[4.6, )
45	[0, 2.8)	[2.8, 4.8)	[4.8, )
46	[0, 2.9)	[2.9, 5.0)	[5.0, )
47	[0, 3.0)	[3.0, 5.2)	[5.2, )
48	[0, 3.1)	[3.1, 5.4)	[5.4, )
49	[0, 3.2)	[3.2, 5.6)	[5.6, )
50	[0, 3.3)	[3.3, 5.8)	[5.8, )
51	[0, 3.4)	[3.4, 6.0)	[6.0, )
52	[0, 3.5)	[3.5, 6.2)	[6.2, )
53	[0, 3.7)	[3.7, 6.4)	[6.4, )
54	[0, 3.8)	[3.8, 6.7)	[6.7, )
55	[0, 4.0)	[4.0, 7.0)	[7.0, )
56	[0, 4.2)	[4.2, 7.2)	[7.2, )
57	[0, 4.4)	[4.4, 7.6)	[7.6, )
58	[0, 4.6)	[4.6, 7.9)	[7.9, )
59	[0, 4.8)	[4.8, 8.3)	[8.3, )
60	[0, 5.0)	[5.0, 8.6)	[8.6, )
61	[0, 5.1)	[5.1, 8.9)	[8.9, )
62	[0, 5.3)	[5.3, 9.2)	[9.2, )
63	[0, 5.5)	[5.5, 9.5)	[9.5, )
64	[0, 5.6)	[5.6, 9.7)	[9.7, )
65	[0, 5.7)	[5.7, 9.9)	[9.9, )
66	[0, 5.8)	[5.8, 10.0)	[10.0, )
67	[0, 5.8)	[5.8, 10.0)	[10.0, )
68	[0, 5.8)	[5.8, 10.0)	[10.0, )
69	[0, 5.7)	[5.7, 10.0)	[10.0, )

10-year absolute risk categories of average risk, higher than average risk, and high risk are set to be equal to remaining lifetime risk from age 30 to 80 years of <15%, 15% to <25%, and ≥25%, respectively.

**Table 2 jpm-11-00916-t002:** Age-specific relative risks associated with age-independent and age-dependent absolute risk thresholds for the high-risk category compared to the population average absolute risk, using remaining lifetime risk and 10-year absolute risk for breast cancer in women in Canada (2012–2016).

	**Remaining Lifetime Risk Metric**				
		Age-independent risk threshold		Age-dependent risk threshold	
Current age	Population average remaining lifetime risk %	Remaining lifetime risk threshold for high risk (%)	Relative risk (high risk vs. population average risk)	Remaining lifetime risk threshold for high risk (%)	Relative risk (high risk vs. population average risk)
30	10.1	25.0	2.7	25.0	2.7
35	10.0	25.0	2.7	24.8	2.7
40	9.8	25.0	2.8	24.3	2.7
45	9.3	25.0	2.9	23.2	2.7
50	8.6	25.0	3.2	21.6	2.7
55	7.7	25.0	3.6	19.5	2.7
60	6.7	25.0	4.1	17.1	2.7
65	5.4	25.0	5.2	13.9	2.7
	**10-year Absolute Risk Metric**				
		Age-independent risk threshold		Age-dependent risk threshold	
Current age	Population average 10-year absolute risk %	10-year absolute risk threshold for high risk (%)	Relative risk (high risk vs. population average risk)	10-year absolute risk threshold for high risk (%)	Relative risk (high risk vs. population average risk)
30	0.4	1.1	2.7	1.1	2.7
35	0.8	1.1	1.4	2.2	2.7
40	1.3	1.1	0.8	3.6	2.7
45	1.8	1.1	0.6	4.8	2.7
50	2.2	1.1	0.5	5.8	2.7
55	2.6	1.1	0.4	7.0	2.7
60	3.3	1.1	0.3	8.6	2.7
65	3.8	1.1	0.3	9.9	2.7

Relative risks calculated using Equation (1). High risk is defined as remaining lifetime risk (from age 30 to 80 years) of 25%.

## Data Availability

The data underlying this article are available in the article and in its online Appendix A.

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
