# Peer review of "Should Age-Dependent Absolute Risk Thresholds Be Used for Risk Stratification in Risk-Stratified Breast Cancer Screening?"

_jpm, 2021, doi:10.3390/jpm11090916_

Round 1

Reviewer 1 Report

This is a well written study using the age-dependent as a risk stratification for breast cancer screening. 

1) In the Introduction - please expand why was this alternative approach done ? what are the advantages ? How does it challenge the existing paradigms ?

2) discussion needs to be more fleshed out, do the results conflict with those of other researches and have potential explanations been proposed?

3) Any potential limitations?

4) Strength of the study ? Cost effective as screening will reduce in some pts.

Author Response

Reviewer 1:

 This is a well written study using the age-dependent as a risk stratification for breast cancer screening.

Thank you

1) In the Introduction - please expand why was this alternative approach done? what are the advantages ? How does it challenge the existing paradigms ?

We have added the rationale for using age-varying risk threshold on p3 para 2. The advantages or the implications are shown in the Results section.

2) discussion needs to be more fleshed out, do the results conflict with those of other researches and have potential explanations been proposed?

We have added further clarifications in the Discussion section, para1, 3, 4, 5, and in the Results p7 last paragraph.

This issue of varying the absolute risk threshold by age is not explicitly discussed in the published literature.  We have cited the UK NICE guidelines that use age-specific risk thresholds. Recently, the European Society of Cardiology published its guidelines https://www.escardio.org/Guidelines/Clinical-Practice-Guidelines/2021-ESC-Guidelines-on-cardiovascular-disease-prevention-in-clinical-practice and for the first time, they explicitly use different absolute risk thresholds for different age groups. However, they do not explain, as we do in our manuscript, why to use age-specific cut-off points for risk stratification.

As we have pointed in the Introduction, the US and the Canadian guidelines for breast cancer use the term ‘lifetime risk’ when in fact ‘remaining lifetime risk’ is what is calculated.  For the former, while we can use age-constant risk threshold, for the latter we need age-varying risk threshold as remaining lifetime risk varies by age. 

3) Any potential limitations?

Indeed, we have pointed the limitation is absence of empirical evidence that age-dependent absolute risk threshold that optimizes the benefit to harm ratio of risk-stratified screening.

4) Strength of the study ? Cost effective as screening will reduce in some pts.

We have added the potential benefit of improving the cost-effectiveness of the screening programme sentence (p9 last paragraph).

Reviewer 2 Report

It is well written article and identifies age-dependent  absolute risk threshold for risk stratification. 

Abstract: Remove 'In turn'

Page 5. Last line. Correct Figure 1 

Author Response

Reviewer 2:

It is well written article and identifies age-dependent  absolute risk threshold for risk stratification. 

Thank you

Abstract: Remove 'In turn'

Done

Page 5. Last line. Correct Figure 1 

Thank you, we have clarified Figure 1 legend